# Importance of Personality for Objective and Subjective-Physical Health in Older Men and Women

**DOI:** 10.3390/ijerph17238809

**Published:** 2020-11-27

**Authors:** Teresa Montoliu, Vanesa Hidalgo, Alicia Salvador

**Affiliations:** 1Laboratory of Social Cognitive Neuroscience, Department of Psychobiology and IDOCAL, Faculty of Psychology, University of Valencia, 46010 Valencia, Spain; teresa.montoliu@uv.es (T.M.); alicia.salvador@uv.es (A.S.); 2IIS Aragón, Department of Psychology and Sociology, Area of Psychobiology, University of Zaragoza, 44003 Teruel, Spain

**Keywords:** neuroticism, conscientiousness, extraversion, metabolic syndrome, subjective health, older people, gender

## Abstract

Objective and subjective health generally have a positive relationship, although their association may be moderated by factors such as gender and personality. We aimed to analyze the association between personality and objective (metabolic syndrome (MetS)) and subjective-physical health in older men and women. For this purpose, in 138 participants (53.6% women, Mage = 66.85), neuroticism, conscientiousness, extraversion, openness, and agreeableness (NEO Five Factor Inventory), subjective-physical health (Short Form Health Survey, SF-36), and MetS (employing waist circumference, blood pressure, triglycerides, high-density lipoprotein cholesterol, and glycated hemoglobin) were assessed. Logistic regression analysis was performed to analyze whether personality was associated with MetS. Moreover, hierarchical regression analyses were conducted to analyze the relationship between personality or MetS, and subjective-physical health. Finally, gender and personality moderation analyses were performed with PROCESS. Results showed that higher neuroticism was associated with an increased likelihood of MetS, whereas higher neuroticism and lower extraversion were associated with lower subjective-physical health. Moreover, the negative relationship between MetS and subjective-physical health was stronger in individuals with low conscientiousness. Regarding gender differences, only in women, higher extraversion was related to a decreased likelihood of MetS, and MetS was related to lower subjective-physical health. In conclusion, higher neuroticism is the main vulnerability health factor, whereas to a lesser extent extraversion and conscientiousness are protective factors. Furthermore, the association between objective and subjective health is not direct, but it may vary depending on personality and gender.

## 1. Introduction

Personality reflects individual differences in relatively enduring patterns of thoughts, feelings, and behaviors. There is a general consensus that the big five personality traits represent the most basic adult personality factors [1]. These five traits, which indicate patterns and trends, are the following: neuroticism (being emotionally unstable and experiencing negative emotions); extraversion (being assertive and social, experiencing positive affect, and seeking excitement); openness (being creative, curious, sensitive to aesthetics, and open to new ideas and experiences); agreeableness (being altruistic, trusting, modest, and compliant); and conscientiousness (being persistent, organized, and goal-directed, and showing self-control and self-discipline) [1,2]. Personality traits have been associated with subjective well-being, health, and mortality risk during aging [3,4,5]. In addition, some authors have suggested that the link between personality traits and health may be cumulative over time, and, therefore, personality could have a greater influence on disease in old age [5].

Personality has been considered a risk factor for metabolic syndrome (MetS) [6], whose prevalence rises with age [7] and increases the risk of cardiovascular outcomes and all-cause mortality [8]. According to the National Cholesterol Education Program Adult Treatment Panel III (NCEP-ATP III), MetS is diagnosed when an individual has three or more risk factors, including high waist-circumference (WC), high blood pressure, high triglycerides (TGL) and low high-density lipoprotein cholesterol levels (HDL-cholesterol), and high glucose [9]. Sutin et al. [10,11] analyzed the relationship between the big five personality traits and metabolic health, and they concluded that the association between neuroticism and conscientiousness and metabolic dysfunction starts early, whereas the association with agreeableness emerges at older ages, suggesting that this association may unfold across the lifespan [11]. Furthermore, they observed that gender moderated the associations between impulsiveness (a facet of neuroticism), openness, and MetS [10]. Therefore, these results suggest an association between some personality traits and MetS in older adults, as well as gender differences in these associations.

Personality has also been related to subjective health (also called self-assessed health, or health-related quality of life [HRQoL]), and this association appears to be more pronounced with age [12]. In older adults, subjective health has been associated with health-literacy, self-efficacy, physical health-promoting behavior, perceived emotional-informational support [13], and health outcomes and mortality [14]. Subjective health has been widely assessed with the Short Form Health Survey (SF-36), which contains eight subscales that can be grouped into the two main components, subjective physical and mental health [15]. In studies that analyzed the association between personality traits and the subjective-physical health scale or some subscale components, higher neuroticism was related to worse subjective-physical health in most studies [4,12,16,17,18], but not in all of them [19,20]. Similarly, higher extraversion [4,12,16,17,20] and conscientiousness [16,17,18,19,20] were related to better subjective-physical health in most studies. In contrast, openness [4,12,20] and agreeableness [4] were associated with subjective-physical health subscales in a few studies. In addition, only Jerram and Coleman [4] analyzed these associations separately in men and women, and they suggested that the results for the whole sample were a poor reflection of the results for men and women separately. Therefore, the study of the moderating effect of gender is a pending issue. 

Although objective health is usually positively associated with subjective health, subjective health cannot be considered a simple reflection of physical health. In addition, this association tends to be weaker with age. Older adults may modify their criteria for perceived health to emphasize deteriorations in physical health, and they are more likely to base their physical health on attitudes (i.e., a positive outlook on life) and behaviors (i.e., a healthy lifestyle) [21]. A recent study observed that personality moderated the association between objective (including clinical [MetS risk factors: hypertension, glucose and Glycated Hemoglobin (HbA1c)], motor and cognitive status) and subjective (measured with a single item) health in older adults with type 2 diabetes [22]. Elran-Barak, Weinstein, Beeri, and Ravona-Springer [22] observed that objective–subjective health associations were stronger in individuals with an “unfavorable” personality, specifically those with high neuroticism and low openness and agreeableness. Moreover, regarding the MetS components, they reported that the association between hypertension and subjective health was only observed in individuals with high neuroticism [22]. The role of personality in the association between objective (MetS) and subjective health has hardly been analyzed in older adults. Moreover, some studies suggest that the association between MetS and subjective health is only observed in women [23].

Hence, the present study aimed to analyze the relationships between the personality traits and both objective (MetS) and subjective-physical health in older people, as well as gender differences in these relationships. In addition, we aimed to examine the moderating role of personality and gender in the association between objective and subjective-physical health. We hypothesized that individuals with higher neuroticism and lower conscientiousness and agreeableness would be more likely to have MetS [10]. We also hypothesized that higher neuroticism and lower conscientiousness would be related to lower subjective-physical health [18]. Finally, we expected neuroticism to moderate the association between MetS and subjective-physical health, and so a negative association would be observed when neuroticism is high [22]. Because gender differences in these associations have seldom been explored, we did not formulate specific hypotheses. 

## 2. Materials and Methods

### 2.1. Participants

One-hundred and thirty-eight participants (64 men and 74 women), ranging in age from 56 to 80 years (M = 66.85, SD = 4.97), were recruited from a study program at the University of Valencia (Spain) for people over 55 years old. This study is part of the MNEME Project, a large research project designed to explore the relationships among cognitive functioning and psychological and physiological factors, including hypothalamic–pituitary–adrenal axis functioning, in older people. Exclusion criteria for this research were: smoking more than 10 cigarettes a day, alcohol or other drug abuse, diabetes, or severe psychiatric or endocrine diseases. All the women were postmenopausal, and none of them were receiving hormone replacement therapy. None of the participants scored less than 27 on the Spanish version of the Mini-Mental Status Examination [24], indicating the absence of cognitive impairment. Of the total sample, the five MetS risk factors were obtained from only 98 participants, who were categorized according to the presence or absence of MetS (see Section 2.4). 

### 2.2. Procedure

The study was conducted in accordance with the Declaration of Helsinki, and the protocol was approved by the Research Ethics Committee of the University of Valencia (Principal Investigator: Prof. Dr. Alicia Salvador). All participants provided written informed consent.

Participants were asked to attend one session that took place at 10:00 or 12:00 h in the Laboratory of Social Cognitive Neuroscience at the University of Valencia. Once in the laboratory, participants filled out the questionnaires to measure their personality traits and subjective-physical health (see Section 2.3 and Section 2.5). In addition, WC, blood pressure, TGL, HDL-cholesterol, and HbA1c were measured to assess MetS. Participants were also asked for their educational level and current degree of physical activity. To do so, participants had to check one of the following options: (i) “Avoid walking or getting tired. Always use the elevator and means of transport” (avoid physical activity); (ii) “Walk for pleasure, use the stairs. Occasionally do exercises that cause sweating or fast breathing” (light physical activity); (iii) “regularly participate in physical activities that require moderate efforts such as maintenance gymnastics, biking, walking, dancing, yoga, golf, ping-pong, horseback riding, or similar activities” (moderate physical activity); or (iv) “regularly participate in vigorous physical activities, such as running, swimming, brisk walking, rowing, cycling, playing tennis, basketball, or similar activities” (vigorous physical activity). 

### 2.3. Personality Traits

The Spanish version [25] of the NEO Five Factor Inventory (NEO-FFI) [1] was used to measure the personality traits. The NEO-FFI consists of 60 items that measure the big five personality traits (neuroticism, extraversion, openness, agreeableness, and conscientiousness), with 12 items for each. The items are answered on 5-point scales, and higher scores indicate a higher degree of the trait. The internal reliabilities for the subscales in the present study were good, with Cronbach’s alphas of 0.81 (neuroticism), 0.84 (extraversion), 0.66 (openness), 0.73 (agreeableness), and 0.83 (conscientiousness).

### 2.4. Objective Health: MetS

WC (cm) was measured to assess central adiposity. Blood pressure (mm Hg) was assessed using the ORMON M6W automatic blood pressure monitor (ORMON Healthcare, Kyoto, Japan). Participants were in a seated position, and three measures were taken at 30-s intervals and averaged. TGL, HDL-cholesterol, and HbA1c were assessed from capillary blood samples with the Cobas b 101 system (Roche Diagnostics, Barcelona, Spain). In addition, participants reported whether they took medication for hypertension or dyslipidemia. 

MetS was defined as the presence of three or more of the following risk factors: (1) elevated WC (≥102 cm in men and ≥88 cm in women), (2) elevated blood pressure (≥130 mmHg systolic blood pressure, ≥85 mmHg diastolic blood pressure, or on antihypertensive drug treatment), (3) elevated TGL (≥150 mg/dL or drug treatment for elevated TGL), (4) reduced HDL-cholesterol (<40 mg/dL in men and <50 mg/dL in women, or drug treatment for reduced HDL-cholesterol), and (5) elevated HbA1c (≥5.7%). We followed the guidelines from the US NCEP-ATP III [9], except for the hyperglycemia risk factor, which is defined as elevated fasting glucose ≥100 mmol/L, which corresponds to a prediabetes condition. Because we did not obtain fasting glucose, we considered ≥5.7% HbA1c, which is the cut-point for prediabetes [26], to identify the hyperglycemia risk factor. Participants who showed 3 or more risk factors were given a score of 1 on the category of presence of MetS, and the rest were given a score of 0. 

### 2.5. Subjective-Physical Health

The Spanish version [27] of the Short Form Health Survey (SF-36) [15] was administered to measure subjective health. It consists of 36 items distributed in eight subscales: physical functioning (PF), role-physical (RF), bodily pain (BP), general health (GH), vitality (V), social functioning (SF), role-emotional (RE), and mental health (MH). For each subscale, the items were transformed into a scale ranging from 0 (worst health) to 100 (best health), using the algorithms and instructions provided in the manual. The eight subscales can be grouped in two summary measures: physical health (PF, RF, BP, GH, and V) and mental health (GH, V, SF, RE, and MH) scales. Based on the aims of the present study, we only employed the physical health scale, which had a Cronbach’s α = 0.71. 

### 2.6. Statistical Analysis

Participants’ characteristics were described using descriptive statistic tools (means, standard deviations, (SD) or percentages) when appropriate, for the total sample and for men and women independently. To investigate gender differences in age, personality traits, and subjective-physical health, independent sample Student-t tests were performed. Pearson’s Chi-square test was used to assess gender differences in the educational level, physical activity, and MetS. 

To investigate whether personality traits were associated with MetS adjusted for covariates (age, gender, educational level, and physical activity), logistic regression analysis was performed, including MetS as dependent variable, the covariates in block one, and the personality traits in block two. Then, to determine the relationships between personality traits and subjective-physical health, we performed hierarchical regression analysis, including subjective-physical health as the dependent variable, the covariates in block one, and the personality traits in block two. After that, to analyze the association between MetS and subjective-physical health, we again performed hierarchical regression analysis, including subjective-physical health as the dependent variable, the covariates in block one, and the MetS in block two. Finally, we analyzed whether gender or personality traits moderated these relationships, using the PROCESS macro in SPSS (v3.4) (Model 1) with 5000 bootstrapped samples. 

Data on NEO-FFI were obtained for the entire sample (N = 138), but there was one missing value for SF-36 (N = 137), and the five MetS risk factors were available for only 98 participants, as mentioned above. Moreover, five multivariate outliers were detected with standardized residuals (±3 SD) and excluded from the analyses. Therefore, the number of participants varied in the different analyses. 

To perform these statistical analyses, version 25.0 of SPSS was used. All *p* values were two-tailed, and the level of significance was taken as *p* < 0.05.

## 3. Results

### 3.1. Participant Characteristics

No significant differences in age, physical activity, neuroticism, openness, or conscientiousness scores were observed between men and women (all *p* ≥ 0.199). However, women showed significantly higher scores on agreeableness (*t* (136) = 3.622, *p* ≤ 0.001) and marginally higher scores on extraversion (*t* (136) = 1.889, *p* = 0.061) than men. Men showed higher educational levels (*X*^2^ (5) = 18.26, *p* = 0.003) and higher subjective-physical health scores (*t* (135) = −3.490, *p* = 0.001) than women (Table 1).

### 3.2. Relationship between Personality and Objective Health (MetS)

Results showed that higher neuroticism was related to an increased probability of MetS (odds ratio (OR) = 1.16, B = 0.150, SE = 0.060, *p* = 0.013), whereas higher agreeableness was only marginally related to a decreased probability of MetS (OR = 0.903, B = −0.102, SE = 0.060, *p* = 0.086). The rest of the personality traits were not associated with MetS (all *p* ≥ 0.347) (Table 2).

Furthermore, gender significantly moderated the association between extraversion and MetS (B = 1.289, SE = 0.527, *p* = 0.014, CI 95% = 0.256, 2.321). Specifically, higher extraversion was negatively related to MetS only in women (B = −0.985, SE = 0.430, *p*= 0.022, CI 95% = −1.826, −0.143). Moreover, gender marginally moderated the association between neuroticism and MetS (B = −1.080, SE = 0.607, *p* = 0.075, CI 95% = −2.271, 0.110). Thus, higher neuroticism was significantly and positively related to MetS only in women (B = 1.480, SE = 0.501, *p* = 0.003, CI 95% = 0.497, 2.463). Gender did not moderate the association between the other personality traits and MetS (all CI 95% included zero) (Table 3).

### 3.3. Relationship between Personality and Subjective-Physical Health

Results showed that neuroticism was negatively related to subjective-physical health (B = −3.017, SE = 0.811, *p* ≤ 0.001, CI 95% = −4.623, −1.412), whereas extraversion was positively related to it (B = 1.833, SE = 0.763, *p* = 0.018, CI 95% = 0.322, 3.343). The rest of the personality traits showed no associations with subjective-physical health (all *p* ≥ 0.141). Specifically, all the predictor variables together accounted for 32% of the variance, and after including neuroticism and extraversion in the model, the amount of variance explained increased to 21% (Table 4).

Gender only marginally moderated the association between conscientiousness and subjective-physical health (∆*R*^2^ = 0.03, B = 0.260, SE = 0.135, *p* = 0.056, CI 95% = −0.007, 0.526). Specifically, higher conscientiousness was positively related to subjective-physical health only in men (B = 0.203, SE = 0.095, *p* = 0.034, CI 95% = 0.016, 0.391). Gender did not moderate the association between the rest of the personality traits and subjective-physical health (all CI 95% included zero) (Table 5). 

### 3.4. Relationship between Objective (MetS) and Subjective-Physical Health

MetS was negatively related to subjective-physical health (B = −22.942, SE = 10.430, *p* = 0.031, 95% CI = −43.676, −2.207). All the predictor variables together accounted for 12% of the variance, and after including MetS in the model, the amount of variance explained increased to 5% (Table 6).

In addition, gender also moderated the association between MetS and subjective-physical health (∆*R*^2^ = 0.05, B = 0.649, SE = 0.299, *p* = 0.033, CI 95% = 0.055, 1.244). Specifically, MetS was negatively related to subjective-physical health only in women (B = −0.706, SE = 0.226, *p* = 0.002, CI 95% = −1.155, −0.256). Moreover, conscientiousness also moderated this association (∆*R*^2^ = 0.04, B = 0.330, SE = 0.163, *p* = 0.046, CI 95% = 0.006, 0.654). Specifically, MetS was negatively related to subjective-physical health in individuals with low (−1SD) (B = −0.674, SE = 0.229, *p* = 0.004, CI 95% = −1.129, −0.219) and medium (B = −.371, SE = 0.159, *p* = 0.022, CI 95% = -0.688, −0.054), but not high (+1SD) (B = −0.067, SE = 0.208, *p* = 0.747, CI 95% = −0.481, 0.346) conscientiousness scores. The rest of the personality traits did not moderate this association (all CI 95% included zero) (Table 7). 

## 4. Discussion

This study aimed to delve into the relationships between objective and subjective-physical health and several personality traits in older people. Our results showed several important associations, particularly related to neuroticism and extraversion. Specifically, higher neuroticism was associated with an increased likelihood of MetS and lower subjective-physical health. Furthermore, extraversion was positively related to subjective-physical health. Moreover, MetS was negatively related to subjective-physical health, and conscientiousness moderated this association. Finally, gender moderated the association between extraversion and likelihood of MetS, and between MetS and subjective-physical health. 

### 4.1. Personality and Objective Health (MetS)

Mommersteeg and Pouwer [6], in a systematic review, reported that although the results for the association between personality and MetS were inconclusive, all the significant findings pointed to an increased risk of higher MetS prevalence in a more distressed, angry, or hostile personality. In line with this study, our results reported that older individuals with higher neuroticism scores showed an increased likelihood of having MetS. Moreover, Sutin et al. [10] reported that in adults over 45 years old, high neuroticism and low agreeableness and conscientiousness were associated with MetS. In line with this study, we observed that high neuroticism was associated with MetS. However, contrary to what we expected, agreeableness was only marginally related to MetS, and we failed to observe an association between conscientiousness and MetS. These discrepancies might be explained by the different age ranges of the participants studied and the different percentages of MetS incidence in our study (over 56 years, 62.2% with MetS) and in Sutin et al.’s [10] study (over 45 years, 19% with MetS). 

Furthermore, we observed that gender moderated the association between extraversion and MetS. Specifically, we observed that higher extraversion was associated with a decreased likelihood of having MetS only in women. Although Sutin et al. [10] did not observe an association between extraversion and MetS, other study found that higher extraversion (measured with the Japanese version of the Eysenck Personality Questionnaire revised short form) was related to greater MetS components in adults over 30 years old (18.2% with MetS) [28]. Previous research has reported that extraversion was related to higher WC or Body Mass Index (BMI), both indicators of visceral obesity, which is a risk factor for MetS, in men, whereas extraversion was related to a lower BMI in women [11,29,30,31]. Therefore, this evidence, along with our results, suggests that higher extraversion could protect against MetS only in women. Moreover, previous research also showed that the positive association between neuroticism and WC or BMI was only observed in women, or it was stronger in women than in men [11,29,30,31]. Although our results showed that gender only marginally moderated the association between neuroticism and MetS, we observed that higher neuroticism was associated with an increased likelihood of having MetS only in women. Therefore, our results suggest that higher neuroticism could have a detrimental effect on MetS mainly in women.

### 4.2. Personality and Subjective-Physical Health

Our results showed that higher neuroticism and lower extraversion were related to worse subjective-physical health. To our knowledge, only one other study considered the main physical health scale of the SF-36 questionnaire [18]. Similar to our findings, in healthy older adults, Löckenhoff et al. [18] reported that higher neuroticism was related to worse subjective-physical health, although they did not find an association between extraversion and subjective-physical health. However, in line with our results, other studies also observed that higher extraversion was related to higher scores on several independent SF-36 physical subscales, specifically on general health [12,16], physical functioning [17], physical role [20], and vitality [4]. Löckenhoff et al. [18] also reported that higher conscientiousness was related to better subjective-physical health. Although we did not observe this association in the entire sample, we found that gender marginally moderated the association between conscientiousness and subjective-physical health. Specifically, higher conscientiousness was related to better subjective-physical health only in men. Similar to our findings, Jerram and Coleman [4] also reported that conscientiousness was related to better subjective-physical health in men (general health and vitality subscales), but to worse subjective-physical health in women (physical function subscale). Therefore, this evidence, along with our results, suggests that conscientiousness could be protective of subjective-physical health only in men. 

### 4.3. Relationship between Objective (MetS) and Subjective-Physical Health: Moderating Role of Personality

As expected, objective health (presence of MetS) was related to worse subjective-physical health, although this association was only observed in women [23]. We did not observe gender differences in objective health (MetS), but women reported lower subjective-physical health, as in other studies [22,32,33]. Therefore, one possible reason for this is that women tend to be more tuned in to their emotions and use less effective coping strategies than men [32,33,34]. Furthermore, we observed that the association between objective and subjective-physical health was moderated by conscientiousness, which in turn has been related to coping strategy selection [35]. Specifically, the presence of MetS was related to worse subjective-physical health in individuals with low and medium, but not high, conscientiousness scores. We expected to observe this association in individuals with higher neuroticism, as reported by Elran-Barak et al. [22] in older adults with type 2 diabetes. However, although in that study conscientiousness did not moderate any objective-subjective health association, they reported that subjective health was related to several objective measures (hypertension, Hba1c, motor, and cognitive status) only in the individuals with low conscientiousness, in line with our results. Therefore, although our results do not exactly match those of Elran-Barak et al. [22], they support the idea that associations between objective and subjective health tend to be stronger in older people with unfavorable personality traits. 

### 4.4. Personality and Health in Older People: General Conclusions

Of the big five personality traits, neuroticism and conscientiousness have been more consistently related to health in previous literature. Individuals with higher neuroticism are more likely to engage in risky health behaviors (smoking, alcohol consumption, less physical exercise, and an unhealthy diet), and this personality trait has been considered a robust predictor of different health disorders and mortality [36,37]. On the contrary, conscientiousness has been strongly associated with healthier behaviors and longevity [36]. However, the relationship between extraversion and health is less clear, perhaps because this personality trait has been linked to both positive (diet and exercise) and negative (alcohol and smoking) health behaviors [38]. Apart from health behaviors, emotional regulation and coping strategies could also help to explain the association between personality and objective-subjective health in older adults. Although aging is related to loss and poor health outcomes, older people often report higher levels of well-being than younger adults. One possible explanation is that older adults show enhanced emotion regulation [39], which has been related to personality traits [40]. In a recent meta-analysis, Barańczuk [40] observed that lower levels of neuroticism, to a greater degree, but also higher levels of extraversion, openness, agreeableness, and conscientiousness, were associated with more adaptive and fewer maladaptive emotion regulation strategies. Moreover, in a meta-analysis, Connor-Smith and Flachsbart [41] showed that all the big five personality traits were linked to coping, but particularly neuroticism, extraversion, and conscientiousness. 

In line with this background, our findings support previous evidence, because neuroticism emerged as the strongest risk factor for both objective and subjective-physical health in older people [37]. Moreover, our research also shows that extraversion is a protective factor for both subjective and objective health, at least in women. Although extraversion has been linked to both positive and negative health behaviors [38], this personality trait has been associated with more positive affect, well-being [42,43], and adaptive coping [41] and emotion regulation strategies [40]. However, although conscientiousness has been strongly associated with healthier behaviors and better coping and emotion regulation abilities [36], we failed to observe an association between this personality trait and objective or subjective-physical health. We only observed that high levels of conscientiousness were related to better subjective-physical health in men, but this gender moderation was marginal. Nevertheless, conscientiousness modified the association between objective and subjective-physical health. Therefore, although to a lesser degree than expected, our study also confirms that conscientiousness is a protective factor for health in older people. Our results also reveal that agreeableness and openness are not related to objective health, at least not to MetS, or subjective-physical health in healthy older adults. Finally, these results also show that gender is an important factor to consider when analyzing the relationship between personality and health, which could help to explain mixed previous findings. 

Our study has some limitations. First, due to the correlational nature of the results, we cannot claim causal relationships. In addition, the sample is characterized by having a generally good psychological and physical health condition, and therefore the results cannot be generalized to all older people. However, this has the advantage of reducing the possible effects of confounding variables. Furthermore, in this sample, participants ranged from 56 to 80 years old, whereas the cutoff to refer to older persons is usually at least 65 years old [44]. Moreover, other possible confounding variables, such as sleep [45,46] and diet [47,48], were not measured in this research and should be included in future studies. Finally, because the magnitude of the results obtained was generally small (*OR* and ∆*R*^2^), further studies with larger sample sizes would make it possible to confirm them. Despite these limitations, our study also has several strengths, such as the simultaneous measurement of objective (MetS) and subjective-physical health with a multiple-item questionnaire and the gender moderation analysis in a sample of older people. 

## 5. Conclusions

Our results confirm that, also in healthy older adults, higher neuroticism is the main vulnerability factor for both objective (MetS) and subjective-physical health. By contrast, to a lesser extent, extraversion and conscientiousness are protective factors for objective and/or subjective-physical health. Furthermore, objective and subjective health are not directly related, but they may vary depending on personality traits and gender. Conscientiousness may affect the way individuals perceive their health, and men and women may perceive their health differently, with possible implications for well-being and health. 

## Figures and Tables

**Table 1 ijerph-17-08809-t001:** Characteristics of the total sample and for men and women.

	Total	Men	Women	*p*
N (%)	138	64 (46.4%)	74 (53.6%)	
Age M (SD)	66.85 (4.97)	67.11 (4.50)	66.62 (5.37)	0.568
Educational level N (%)				0.003
Without studies	1 (0.7%)	0 (0%)	1 (1.4%)	
Primary school	26 (19%)	4 (6.3%)	22 (30.1%)	
Secondary school	28 (20.4%)	13 (20.3%)	15 (20.5%)	
Graduate (3-year degree)	35 (25.5%)	18 (28.1%)	17 (23.3%)	
Graduate (5-year degree)	43 (31.4%)	25 (39.1%)	18 (24.7%)	
Postgraduate	4 (2.9%)	4 (6.3%)	0 (0%)	
Physical activity N (%)				0.226
Avoid	13 (9.6%)	5 (8.1%)	8 (11%)	
Light	31 (23%)	13 (21%)	18 (24.7%)	
Moderate	75 (55.6%)	33 (53.2%)	42 (57.5%)	
Vigorous	16 (11.9%)	11 (17.7%)	5 (6.8%)	
Neuroticism M (SD)	17.06 (6.25)	16.32 (6.20)	17.70 (6.26)	0.199
Extraversion M (SD)	29.10 (6.56)	27.98 (6.65)	30.08 (6.36)	0.061
Openness M (SD)	30.04 (5.14)	29.81 (4.90)	30.24 (5.05)	0.625
Agreeableness M (SD)	31.86 (5.14)	30.23 (4.52)	33.28 (5.25)	≤ 0.001
Conscientiousness M (SD)	32.35 (6.09)	32.50 (6.40)	32.22 (5.85)	0.796
Subjectivephysical health M (SD)	392.67 (63.47)	411.67 (47.76)	376.02 (56.35)	0.001
MetS N (%)	61 (62.2%)	28 (59.6%)	33 (64.7%)	0.601

Note: M = mean; SD = standard deviation; MetS = metabolic syndrome.

**Table 2 ijerph-17-08809-t002:** Relationship between personality traits and metabolic syndrome (MetS).

	OR	CI 95% (OR)	B	SE	*p*
Model 1					
Age	1.168	1.050, 1.300	0.156 **	0.054	0.004
Gender	1.027	0.372, 2.833	0.026	0.518	0.959
Educational level	0.549	0.346, 0.870	−0.600 *	0.235	0.011
Physical activity	0.943	0.505, 1.762	−0.059	0.319	0.853
Model 2					
Neuroticism	1.161 *	1.032, 1.307	0.150 *	0.060	0.013
Extraversion	1.013	0.919, 1.117	0.013	0.050	0.788
Openness	1.020	0.912, 1.141	0.020	0.057	0.723
Agreeableness	0.903 ^#^	0.804, 1.015	−0.102	0.060	0.086
Conscientiousness	1.053	0.946, 1.171	0.051	0.055	0.347

Note: MetS = metabolic syndrome; OR = Odds ratio; CI = Confidence interval; SE = Standard error; ** *p* < 0.01. * *p* < 0.05. ^#^
*p* < 0.10.

**Table 3 ijerph-17-08809-t003:** Gender moderations in the relationship between personality traits and MetS.

	*X* ^2^	*X*^2^ (*p*)	B	SE	*p*	CI 95%
GenderXNeuroticism	3.494	0.062	−1.080 ^#^	0.607	0.075	−2.271, 0.110
Women			1.480 **	0.501	0.003	0.497, 2.463
Men			0.400	0.364	0.272	−0.313, 1.113
GenderXExtraversion	6.870	0.009	1.289 *	0.527	0.014	0.256, 2.321
Women			−0.985 *	0.430	0.022	−1.826, −0.143
Men			0.304	0.337	0.367	−0.357, 0.966
GenderXOpenness	0.743	0.389	0.422	0.492	0.391	−0.541, 1.386
GenderXAgreeableness	1.282	0.257	−0.573	0.511	0.262	−1.574, 0.428
GenderXConscientiousness	1.170	0.279	0.586	0.545	0.283	−0.483, 1.654

Note. MetS = metabolic syndrome; *X*^2^ = Chi-square; SE = Standard error; CI 95% = Confidence interval; ** *p* < 0.01. * *p* < 0.05. ^#^
*p* < 0.10.

**Table 4 ijerph-17-08809-t004:** Relationship between personality traits and subjective-physical health.

	*R* ^2^	∆*R*^2^	B	SE	*p*	CI 95%
Model 1	0.11	0.11 **				
Age			0.223	0.912	0.807	−1.582, 2.027
Gender			14.407	9.316	0.125	−4.033, 32.847
Educational level			4.372	3.923	0.267	−3.393, 12.137
Physical activity			15.709 **	5.569	0.006	4.685, 26.732
Model 2	0.32	0.21 ***				
Neuroticism			−3.017 ***	0.811	≤0.001	−4.623, −1.412
Extraversion			1.833 *	0.763	0.018	0.322, 3.343
Openness			−1.112	0.864	0.201	−2.822, 0.599
Agreeableness			0.571	0.881	0.519	−1.175, 2.316
Conscientiousness			−1.075	0.726	0.141	−2.513, 0.363

Note: ∆*R*^2^ = R square change; SE = Standard error; CI 95% = Confidence interval; *** *p* < 0.001. ** *p* < 0.01. * *p* < 0.05.

**Table 5 ijerph-17-08809-t005:** Gender moderations in the relationship between personality traits and subjective-physical health.

	∆*R*^2^	B	SE	*p*	CI 95%
GenderXNeuroticism	0.00	0.086	0.128	0.503	−0.168, 0.340
GenderXExtraversion	0.01	−0.140	0.133	0.295	−0.403, 0.123
GenderXOpenness	0.02	−0.197	0.138	0.158	−0.470, 0.077
GenderXAgreeableness	0.01	0.142	0.146	0.333	−0.147, 0.431
GenderXConscientiousness	0.03	0.260 ^#^	0.135	0.056	−0.007, 0.526
Women		−0.056	0.096	0.559	−0.246, 0.134
Men		0.203 *	0.095	0.034	0.016, 0.391

Note: ∆*R*^2^ = R square change; SE = Standard error; CI 95% = Confidence interval; * *p* < 0.05. ^#^
*p* < 0.10.

**Table 6 ijerph-17-08809-t006:** Relationship between MetS and subjective-physical health.

	*R* ^2^	∆*R*^2^	B	SE	*p*	CI 95%
Model 1	0.07	0.07				
Age			−0.807	0.984	0.414	−2.762, 1.148
Gender			7.849	10.327	0.449	−12.678, 28.376
Educational level			0.651	4.309	0.880	−7.914, 9.215
Physical activity			11.435 ^#^	6.105	0.064	−0.699, 23.570
Model 2	0.12	0.05 *				
MetS			−22.942 *	10.430	0.031	−43.676, −2.207

Note: MetS = metabolic syndrome; ∆*R*^2^ = R square change; SE = Standard error; CI 95% = Confidence interval; * *p* < 0.05. ^#^
*p* < 0.10.

**Table 7 ijerph-17-08809-t007:** Gender or personality moderation in the relationship between MetS and subjective-physical health.

	∆*R*^2^	B	SE	*p*	CI 95%
MetSXgender	0.05	0.649 *	0.299	0.033	0.055, 1.244
Women		−0.706 **	0.226	0.002	−1.155, −0.256
Men		−0.056	0.214	0.792	−0.481, 0.368
MetSXNeuroticism	0.02	−0.216	0.159	0.180	−0.533, 0.101
MetSXExtraversion	0.01	0.106	0.151	0.485	−0.194, 0.406
MetSXOpenness	0.00	0.066	0.162	0.683	−0.256, 0.389
MetSXAgreeableness	0.00	0.079	0.156	0.613	−0.232, 0.391
MetSXConscientiousness	0.04	0.330 *	0.163	0.046	0.006, 0.654
Low conscientiousness (−1SD)		−0.674 **	0.229	0.004	−1.129, −0.219
Medium conscientiousness (M)		−0.371 *	0.159	0.022	−0.688, −0.054
High conscientiousness (+1SD)		−0.067	0.208	0.747	−0.481, 0.346

Note: MetS = metabolic syndrome; ∆*R*^2^ = R square change; SE = Standard error; CI 95% = Confidence interval; M = mean; SD = standard deviation. ** *p* < 0.01. * *p* < 0.05.

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
