# Peer review of "Importance of Personality for Objective and Subjective-Physical Health in Older Men and Women"

_ijerph, 2020, doi:10.3390/ijerph17238809_

Round 1
Reviewer 1 Report
This interesting study analyzed the association between personality and objective (measured as metabolic syndrome) and subjective-physical health in older men and women. The study built some models and controlled each model for age, sex and education.
First, there are some important confounders that could have been included for further adjustment of each model such as sleep, diet and physical activity. If these confounders were not measured in this study, it would be appropriate to express this issue as part of limitations of study.
Second, the metabolic syndrome as an objective health index is a categorical variable whereas the subjective health index is expressed as continuous variable. As we know, by summarizing five continuous variables such as blood pressure and triglyceride into one categorical variable of metabolic syndrome (yes vs. No), the researcher will lose a lot of information. To solve this problem, I advise the authors to use metabolic syndrome severity score and compare the findings with the use of metabolic syndrome.
Third, have you tried to include the two significant personality traits in table 2 in one model? The same question is applicable for Table 3?
Fourth, please present the findings on section 3.3 as Table 4.
Fifth, please summarize the effect modification of sex in different models as Table 5.
Reviewer 2 Report
This is an interesting work due to its relatively unexplored theme in the elderly population, however, in my opinion it needs improvements in the following sections:
INTRODUCTION:
- It would be interesting to go deeper into the determinants of health-related quality of life in older people.
- Even briefly, it would be good to conceptualize the five personality traits.
- Line 55-60, it is recommended to include more up-to-date references that address this issue using the same health instrument. A highly recommended reference is: https://doi.org/10.1002/ijop.12089
METHOD:
- In many countries, including Spain, an older person is considered to be 65 or over. This should be noted as a limitation of the work and briefly discussed.
- The use of the exclusion criteria employed (smoking, alcohol...) should be justified.
- It would be good to justify the use of the stepwise method, as it has received various criticisms. If this is not possible, some of its drawbacks should be mentioned.
- Annotations are required at the bottom of the table with some acronyms used.
DISCUSSION:
- In my opinion, it would be desirable to comment on placing more emphasis in the discussion on the magnitude of the results obtained (e.g., increase in R2).
- When discussing the general results, it could go further into the possible role that emotion regulation in older people (with its well-known peculiarities) could have on the relationships explored.
Reviewer 3 Report
Authors aimed to analyze the relationships between the big five personality traits and other factors with MetS and subjective-physical health in older people, describing the effect of gender in these relationships. Additionally, authors aimed to examine the role of personality and gender in the association between objective and subjective physical health.
Specific points are highlighted in the .pdf file.
Some issues are related with more detailed information on the regression model building and the Result section.
In general terms the manuscript is of high quality and the topic very important, it is well appreciated the way the authors work on showing the results of the study.

Round 2
Reviewer 2 Report
I believe that the changes made by the authors in response to the suggestions of the three reviewers have markedly improved the quality of their study.
For my part, just one additional comment: it would be good to check typos and punctuation errors, especially in lines 65-70 of the latest version of the manuscript.
Best wishes and congrats to the authors,